# GNSS multipath suppression technology based on postcorrelation and independent component analysis

Jian Xu[1]*, Jicheng Ding[2]

1 School of Computer and Information Engineering, Harbin University of Commerce, Harbin, Heilongjiang, China, 2 College of Automation, Harbin Engineering University, Harbin, Heilongjiang, China

* ufoxj@163.com

## Abstract

In urban and other complex environments, satellite navigation signals are vulnerable to multipath interference, especially medium- and short-delay multipath interference, which seriously affects the positioning accuracy. Aiming at the problem that the effect of traditional methods on medium and short multipath suppression is not obvious and it is difficult to accurately estimate the direction of arrival (DOA) when beamforming is used, a multipath suppression scheme based on a postcorrelation and independent component analysis (ICA) algorithm is proposed in this paper. First, the satellite navigation signal on each array element is despread and demodulated by postcorrelation technology to improve the signal-to-noise ratio (SNR), and then the direct signal component is directly extracted by the ICA algorithm to filter the noise while suppressing the multipath interference. Simulation results show that this method can effectively suppress medium and short multipath interference when the SNR of satellite navigation signals is as low as -30 dB, significantly improve the SNR of navigation messages and reduce the bit error rate, which provides a new solution for global navigation satellite system (GNSS) multipath suppression.

**Data Availability Statement:** All relevant data are within the paper.

**Funding:** Financial support was obtained from Harbin University of Commerce's school-level

## 1. Introduction

The global navigation satellite system (GNSS) is widely used due to its global high-precision positioning capability. However, in complex environments such as urban areas, the signals received by satellite navigation receivers are easily affected during propagation. Reflection, diffraction and scattering on the building surfaces in the radio environment induce undesirable multipath propagation [1]; and the resulting ranging error can reach tens of meters, which seriously affects the positioning accuracy and reliability of a receiver.

Since the path of a reflected signal is always longer than that of a direct signal, the arrival of multipath signals has a certain delay compared to a direct signal. Due to its unique structure, the spread spectrum signal used by the GNSS has a certain multipath suppression ability. That is, when the delay of the multipath component relative to the direct signal exceeds 1.5 chips, its self-correlation coefficient is approximately 0, which will not affect the correlation processing

scientific research project (18XN023). Funders played a role in decision to publish.

**Competing interests:** NO authors have competing interests.

of a receiver. However, for medium- and short-delay multipath signals with delays less than 1.5 chips, especially less than 0.5 chips, the multipath effect will distort the correlation function between the received synthetic signal (direct signal plus its multipath component) and the local reference signal, introduce errors in the measurements of the pseudorange and carrier phase, reduce the signal-to-noise ratio (SNR) of the satellite navigation signal, and then cause positioning error [2].

At present, the following are the main types of multipath suppression methods: 1) Loop mitigation: Loop mitigation improves the structure of the correlator or discriminator, such as the narrow correlator [3], high resolution correlator [4] and multipath estimation delay locked loop (MEDLL) [5], in a receiver tracking loop to weaken the impact of multipath signals. However, such methods can only suppress long delay multipath signals but have no obvious effect on medium and short delay multipath signals. 2) Spatial suppression: These methods are divided into two categories: one category suppresses multipath signals using array signal processing; and the other category reduces the impact of multipath signals through appropriate antenna design, such as a choke antenna, which can reduce multipath signals scattered from the ground or low elevation [6]. 3) Data postprocessing technology: This technology collects and postprocesses the received data to suppress multipath interference and includes the Bayesian method [7] and maximum likelihood estimation algorithm [8]. However, this type of method is based on mathematical statistics, and its solutions are more complex. These methods are mainly used in data postprocessing, which is difficult to apply in real-time positioning. The data processing method also includes the method developed from the time domain repetition period based on the satellite constellation, such as sidereal filtering (SF) [9] and advanced sidereal filtering (ASF) [10], this kind of method has difficulties such as large amount of modeling and complicated correction; and the modeling method based on multipath spatial repeatability such as multipath hemispherical map (MHM) [11], trend surface analysis-based multipath hemispherical map (T-MHM) [12] and single-difference multipath hemispherical map (SD-MHM) [13]. In addition, with the development of neural networks and machine learning, there are many new methods to identify line of sight (LOS) and non line of sight (NLOS) signals, including NLOS multipath classification of GNSS signal correlation output using machine learning [14], machine learning based LOS/NLOS classifier for GNSS shadow matching [15], neural networks based GPS spoofing detection [16].

Since this research is aimed at medium and short delay multipath signals in GNSS signals and positioning requires a certain degree of real-time performance, the array signal processing method is considered to suppress multipath signals so as to improve the positioning accuracy of a navigation receiver.

In traditional array signal processing, prior information, such as the direction of the direct signal or multipath interference, needs to be estimated first [17], and then a multipath interference suppression beamforming algorithm [18] or multipath signal receiving beamforming algorithm [19] is used to process the multipath signal. However, in practical applications, it is difficult to estimate the direction of arrival of a satellite navigation signal due to its weak strength. Therefore, it is difficult to implement this type of method.

In this paper, a postcorrelation combined with independent component analysis (ICA) multipath suppression method is proposed. The ICA algorithm is added to the tracking loop. That is, postcorrelation technology is used to track the satellite navigation signal on each array element to achieve despreading and demodulation. The despreading gain is used to improve the SNR, and then the ICA algorithm is used to directly extract the direct signal component so as to suppress multipath signals without identifying LOS and NLOS signals. Since the direct signal extracted from the synthetic signal contains the gain provided by its multipath component, it does not cause the loss of multipath information. In addition, the extracted direct

signal does not need to go through the process of estimating the direction of arrival before beamforming, and the intermediate error generated in direction of arrival (DOA) estimation will not be introduced.

## 2. Signal model

Assuming that a uniform linear array composed of $M$ isotropic sensors receives GNSS signals from $N$ satellites and outputs $M$ parallel mixed signals, the received signal of an array is as follows:

$$X(t) = AS(t) + n(t) \tag{1}$$

where $X(t) = [x_1(t), x_2(t), \cdots, x_M(t)]^T$ is the received signal of each array element, $S(t) = [s_1(t), s_2(t), \cdots, s_N(t)]^T$ is the source signals, and $n(t)$ is additive white Gaussian noise (AWGN). $A = [a_1, a_2, \cdots, a_N]$ is the array manifold, and $a_n$ is a steering vector of the $n$th channel. According to the spatial relationship between the elements of the uniform linear array, $a_n = [1, e^{i2\pi d \sin\theta_n/\lambda}, \cdots, e^{i2\pi(M-1)d \sin\theta_n/\lambda}]^T, (n = 1, 2 \cdots N)$, where $\theta_n$ is the DOA of an array signal, $d$ is the array element spacing, and $\lambda$ is the wavelength of the received signal.

When the array antenna receives the source signal transmitted by each satellite, if $P$ multipath components are also received at the same time, the received signal $X(t)$ is rewritten as:

$$\hat{X}(t) = a_n s_n(t) + \sum_{p=2}^{P} a_{(n,p)} s_{(n,p)}(t) + n(t), (n = 1, 2 \cdots N) \tag{2}$$

where $a_{(n, p)}$ is the steering vector of the $p$th multipath component in the $n$th source signal, and $s_{(n,p)}(t)$ is the $p$th multipath component in the $n$th source signal. It is assumed that multipath signal $s_{(n,p)}(t)$ delays phase $\varphi_{(n, p)}$ relative to the direct signal, that is, the source signal $s_n(t)$. The attenuation coefficient is $c_{(n, p)}$. Then, $s_{(n,p)}(t) = c_{(n,p)} s_n(t) e^{i\varphi_{(n,p)}}$, and Eq (2) becomes:

$$\hat{X}(t) = a_n s_n(t) + \sum_{p=2}^{P} c_{(n,p)} a_{(n,p)} s_n(t) e^{i\varphi_{(n,p)}} + n(t) \tag{3}$$

The goal of multipath suppression processing is to suppress the multipath component $s_{(n,p)}(t)$ in Eq (2).

## 3. Multipath suppression method

To suppress the multipath component $s_{(n,p)}(t)$ in the array signal, the general method is to estimate the DOAs of the direct component and multipath interference component in the received signal and then use the beamforming algorithm to form a null in the multipath interference direction and form a gain in the direct signal direction according to the estimated DOAs [20]. This method loses the information of multipath components. In addition, because the GNSS signal received by a receiver is weak, the signal is usually submerged in noise, and the SNR in indoor environments will even be lower than -30 dB [21]. This greatly affects the accuracy of DOA estimation and then reduces the multipath suppression effect of subsequent beamforming.

In order to suppress multipath signals in a GNSS weak signal environment, this paper uses postcorrelation technology combined with the ICA algorithm. First, the spread spectrum gain is used to make the GNSS signal have a positive SNR, and then the ICA algorithm is used to extract the direct component from the mixed signal. At the same time, the noise is filtered by the characteristics of the algorithm to further improve the SNR of the direct signal.

### 3.1 Postcorrelation

In array signal processing, multipath suppression can be arranged in the precorrelation or postcorrelation stage of the GNSS receiver [22]. The so-called precorrelation places all the processing of sampled signals before correlation, which is the conventional processing method for spatial signals. In this case, the spread spectrum gain cannot be fully utilized. The postcorrelation technique adds the signal processing algorithm to the tracking loop and places it after the correlation processing. The diagram of the postcorrelation ICA algorithm adopted in this paper is shown in Fig 1.

The GNSS signal received by each antenna array element is downconverted into an intermediate frequency (IF) signal through a radio frequency (RF) front-end and analog-to-digital converter (ADC). First, all available satellite signals are detected through the acquisition unit, and all initial code phases and Doppler frequency shifts are transmitted to the tracking unit. Then, the correlator of the tracking unit despreads and integrates all the available signals received on each array element to obtain the navigation messages of all satellite signals (including direct signals and their multipath components) on each array element and the noise generated in correlation processing. It is equivalent to using an array antenna to directly receive navigation messages containing multipath signals. The resulting signal is shown in Eq (4):

$$\tilde{X}_n(t) = a_n D_n(t) + \sum_{p=2}^{P} c_{(n,p)} a_{(n,p)} D_n(t) e^{i\varphi_{(n,p)}} + n(t) + \tilde{n}(t)$$

$$= \left( a_n + \sum_{p=2}^{P} c_{(n,p)} e^{i\varphi_{(n,p)}} a_{(n,p)} \right) D_n(t) + n(t) + \tilde{n}(t) \tag{4}$$

$$= \hat{a}_n D_n(t) + n(t) + \tilde{n}(t)$$

In the above formula, $\hat{a}_n = a_n + \sum_{p=2}^{P} c_{(n,p)} e^{i\varphi_{(n,p)}} a_{(n,p)}$, which is called the composite steering vector, contains multipath component parameter information [23]. $D_n(t)$ is the navigation message transmitted by the $n$th satellite, $\tilde{X}_n(t)$ is the array signal after correlation processing, and $\tilde{n}(t)$ is the noise generated in the correlation. Since each array element signal contains the

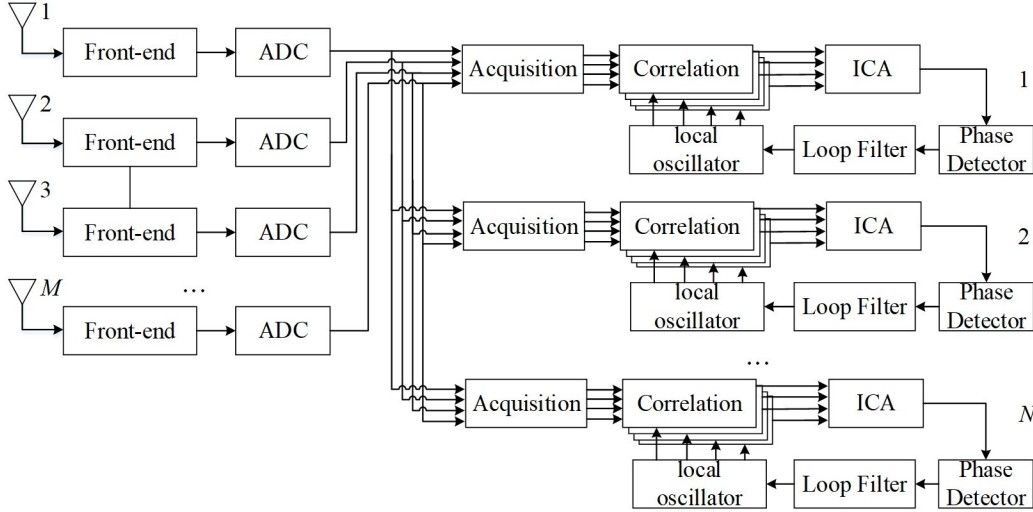

**Fig 1. Diagram of the postcorrelation ICA algorithm for array satellite navigation signals.**

direct and multipath components after satellite signal despreading and the composite steering vector $\hat{\boldsymbol{a}}_n$ is a linear mixture method of these components, the ICA algorithm can be used to extract the direct navigation message $\boldsymbol{D}_n(t)$ after despreading in the tracking loop of each satellite. Then, the direct signal of the satellite continues to be tracked, and the tracking results are fed back to the correlator to form a closed loop.

## 3.2 ICA algorithm

Signals from different sources are usually statistically independent of each other, and the ICA algorithm is used to extract the source signal from the observed mixed signal using this characteristic [24]. In recent years, some multipath mitigation methods using ICA or similar algorithms have also been proposed [25, 26].

The use of the ICA algorithm requires that the statistical characteristics of the source signal meet the following assumptions: 1) All source signals are independent of each other. In the actual environment, this assumption, which is also the basic condition for the adoption of the ICA algorithm, is easy to meet. 2) There can be at most one Gaussian signal in a source signal. Most digital communication signals can be considered non-Gaussian signals, so this assumption holds. 3) The mixed matrix is a full column rank invertible matrix, which ensures that the number of received sensors is not less than the number of source signals. Since there is only a single satellite signal on each tracking loop in the postcorrelation ICA algorithm, the number of source signals is 1, so the condition is met.

The noise-free linear ICA model is able to provide essentially the best possible output signal-to-interference-and-noise ratio (SINR) among all linear transformations of received data in noisy linear models [27]. Therefore, a noise-free linear instantaneous mixed ICA model is discussed here [28]. Assuming that $N$ unknown, statistically independent source signals are transmitted through an unknown linear channel and received by $M$ sensors, the received signal can be expressed as:

$$\boldsymbol{X}(t) = \boldsymbol{A}\boldsymbol{S}(t) \tag{5}$$

ICA is used to process the mixed signal received by the array antenna to estimate the separation matrix $\boldsymbol{W}$ so as to estimate the source signal $\boldsymbol{S}(t)$. The output of ICA model can be expressed as:

$$\boldsymbol{Y}(t) = \boldsymbol{W}\boldsymbol{X}(t) = \boldsymbol{W}\boldsymbol{A}\boldsymbol{S}(t) = \boldsymbol{G}\boldsymbol{S}(t) \tag{6}$$

where $\boldsymbol{Y}(t) = [\boldsymbol{y}_1(t), \boldsymbol{y}_2(t), \cdots, \boldsymbol{y}_n(t)]^{\mathrm{T}}$ is the estimation of unknown source signal $\boldsymbol{S}(t)$, $\boldsymbol{W}$ is the separation matrix and $\boldsymbol{G}$ is the global matrix. As long as there is only one element in each row and column of $\boldsymbol{G}$ close to 1 and the other elements are close to 0, the separation can be considered successful.

There are many methods to solve the ICA problem. Because array signal processing requires complex calculations, the complex fast independent component analysis (cFastICA) algorithm is adopted [29]. The cFastICA algorithm is a fixed point batch ICA algorithm developed on the basis of fast independent component analysis (FastICA) [30]. The core part of the FastICA algorithm is to use negative entropy to measure the degree of mutual statistical independence between the signals to be separated. The algorithm does not need to set a learning rate and has a fast convergence rate. It is one of the most typical algorithms in the ICA algorithm.

In the FastICA algorithm, there are usually two methods to extract independent components: serial orthogonalization and parallel orthogonalization. Serial orthogonalization makes each separation vector orthogonal and estimates the separation vectors individually. Parallel

orthogonalization estimates all separation vectors at the same time, that is, the separation matrix, so that the separation vectors meet the orthogonality at the same time.

The serial orthogonal method can extract the signal of interest step by step. In the postcorrelation ICA algorithm, Eq (4) can be regarded as the navigation message $D_n(t)$ passing through the mixing matrix $\hat{a}_n$ to obtain the received signal $\tilde{X}(t)$. The serial orthogonal method of FastICA can be used to extract the independent component from Eq (4), that is, the direct signal $D_n(t)$, to suppress multipath signals. Since $D_n(t)$ contains the gain provided by its multipath component, it does not cause the loss of multipath information. In addition, since only one independent component is extracted, the process is equivalent to the system noise and the noise generated in the correlation process $n(t) + \tilde{n}(t)$ being filtered out at the same time, which can improve the signal quality.

In order to avoid the complexity of negative entropy calculation, the FastICA algorithm uses a more robust and faster method to approximate the negative entropy, and its cost function is shown in Eq (7):

$$J(\mathbf{y}) \approx [\mathrm{E}\{F(\mathbf{y})\} - \mathrm{E}\{F(\mathbf{v})\}]^2 \tag{7}$$

where $\mathbf{y}$ represents an output variable with zero mean and unit variance, $\mathbf{v}$ represents a Gaussian random variable with zero mean and unit variance, and $F(\cdot)$ represents an arbitrary nonquadratic function. The purpose of the algorithm is to maximize $J(\mathbf{y})$ by selecting the separation matrix $\mathbf{w}$. In the serial orthogonal method, Newton's method is used to solve the optimal solution of the objective function, and the iterative formula of the cFastICA algorithm can be derived as follows:

$$\mathbf{w}_{k+1} = E\{\mathbf{X}(\mathbf{w}_k^H\mathbf{X})^* f(|\mathbf{w}_k^H\mathbf{X}|^2)\} - E\{f(|\mathbf{w}_k^H\mathbf{X}|^2) + |\mathbf{w}_k^H\mathbf{X}|^2 f'(|\mathbf{w}_k^H\mathbf{X}|^2)\}\mathbf{w}_k$$
$$\mathbf{w}_{k+1} = \frac{\mathbf{w}_{k+1}}{\|\mathbf{w}_{k+1}\|} \tag{8}$$

where $\mathbf{w}$ is a row of the separation matrix $\mathbf{W}$ corresponding to a source signal, which is equivalent to $\mathbf{W}$ in this algorithm; $f(\cdot)$ and $f'(\cdot)$ represent the first and second derivatives of $F(\cdot)$, respectively; and $k$ and $k+1$ represent the iterative relationships. After obtaining the separation matrix, according to the principle of Eq (6), the estimation result of the direct navigation message $D_n(t)$ can be obtained:

$$\mathbf{D}_n(t) = \mathbf{w}\tilde{\mathbf{X}}_n(t) \tag{9}$$

In order to improve the convergence characteristics of the ICA algorithm, reduce ill-conditioned problems, eliminate information redundancy and reduce the influence of noise, it is usually necessary to preprocess the received data, including conducting centering and whitening. The centering process subtracts the mean value of the received data so that the mean value of the data after the centering is zero. The whitening process applies a linear transformation to the signal so that each component has unit variance and is not related to other components to obtain the whitened signal. The whitened signal $Z = BX$, where $B$ is the whitening matrix that satisfies the whitened signal $E\{ZZ^T\} = I$ (E represents the mathematical expectation). In practical applications, the whitening matrix $B$ can be derived from the correlation matrix $R$:

$$\mathbf{B} = \mathbf{\Lambda}^{-\frac{1}{2}}\mathbf{U}^T \tag{10}$$

where $\Lambda$ is the eigenvalue matrix and $U$ is the eigenmatrix composed of the eigenvector of $R$. The postcorrelation ICA algorithm needs to use preprocessing to eliminate the influence of

the correlation between the direct signal and its multipath component on the despreading result and then extract the direct signal.

## 4. Simulation and verification

In order to verify the effectiveness and performance, this chapter conducts various simulation verifications on the postcorrelation ICA algorithm shown in Fig 1. The verification contents include the simulation of the improvement effectiveness of the code phase correlation curve and navigation message and the simulation of the performance improvement of the navigation message with the change of the multipath delay and input signal-to-noise ratio.

For the simulation project, we first set the general simulation conditions as shown in Table 1: the second generation Beidou navigation signals from 3 satellites are received by a uniform linear array of 8 array elements; the received SNR is -30 dB; and the incident angles of the direct signals are -40˚, -20˚ and 10˚, respectively. In the space propagation process, due to reflection, the first satellite signal generates two multipath components with incident angles of 30˚ and—10˚, respectively. The signal strength attenuates 0.2 and 0.3 times the direct signal and delays 0.2 and 0.25 chips, respectively; the second satellite signal generates a multipath component with an incident angle of 20˚, the signal strength attenuates 0.1 times the direct signal and delays 0.15 chips; and the third satellite signal does not generate multipath signals.

In urban environment, multipath received signals reflected by different obstacles have different amplitudes and phases. According to whether there is a direct component in the multipath signal, the overall distribution of the final synthesized signal subjects to Rayleigh distribution or Rice distribution [31]. The simulation fitting results shown in Fig 2 show that the simulation parameters in Table 1 are almost consistent with the standard Rice distribution.

### 4.1 Verification of multipath suppression effectiveness

In the GNSS receiver, the received GNSS signal and the local reference signal are correlated and input to the phase detector to obtain the carrier phase error and code phase error. When multipath components exist, the received GNSS signal contains direct components and multipath components. In the correlation processing with the reference signal, in addition to using the correlation between the direct component and the reference signal to obtain the required correlation function, the correlation between the multipath component and the reference signal will also interfere with the output of the receiver correlator and affect the subsequent processing and positioning performance. In a traditional receiver, a delay-locked loop (DLL) generates instant, advance, and delayed reference signals and correlates them with the received signal. The influence of multipath signals on the correlation process can be judged according

**Table 1. General simulation conditions for multipath signals of satellite navigation.**

| Signal type | The second generation | | |
|---|---|---|---|
| | Beidou navigation signals | | |
| Number of satellites | 3 | | |
| Number of elements | 8 | | |
| SNR/dB | -30 | | |
| Incident angles of direct signals | -40˚ | -20˚ | 10˚ |
| Number of multipath components | 2 | 1 | 0 |
| Incident angles of multipath components | 30˚, -10˚ | 20˚ | NA |
| Multipath attenuation/time | 0.2, 0.3 | 0.1 | NA |
| Multipath delay/chip | 0.2, 0.25 | 0.15 | NA |

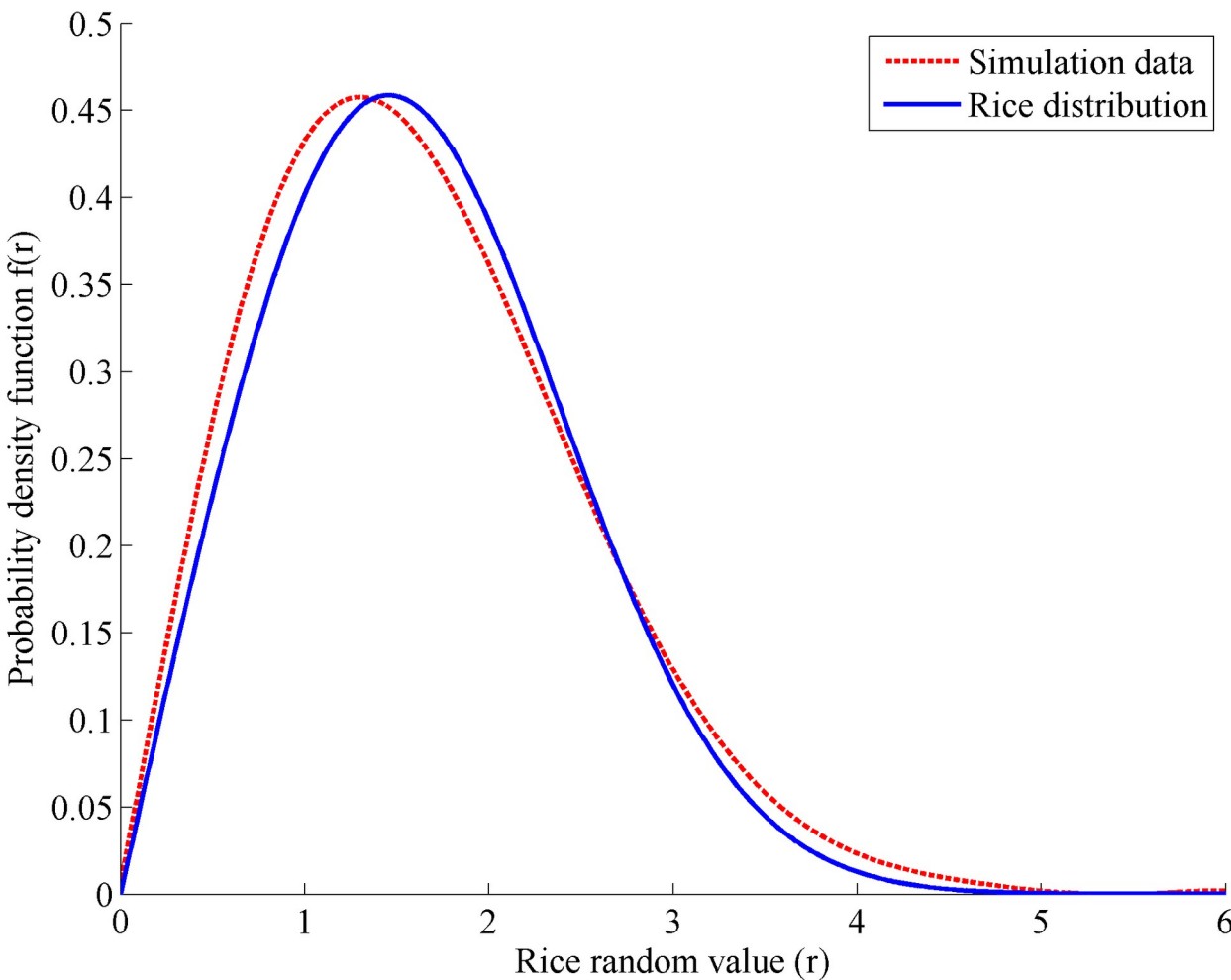

**Fig 2. Fitting of simulation data with standard Rice distribution.**

to the position and shape of the correlation peak. Therefore, the correlation result can be used to establish a code phase curve to verify whether the multipath component has been filtered out of the output signal after ICA processing. Fig 3 shows the comparison of the code phase curves of a satellite navigation signal after being correlated with the local reference signal with and without ICA processing.

As shown in Fig 3, in the presence of a multipath signal, the multipath component has a certain delay relative to the direct component, and the delayed multipath component has a stronger correlation with the delayed reference signal. As a result, the right half of the synthesized code phase curve is distorted, the correlation peak is asymmetric, and the synthesized correlation value has a larger amplitude. After ICA processing, the direct component is extracted, the correlation value between the multipath component and the reference signal is filtered out, and the correlation peak returns to its normal form.

## 4.2 Verification of navigation message improvement effectiveness

As described in Section 3, the postcorrelation ICA algorithm is used to extract the navigation message from a correlated signal by using the serial orthogonal method of FastICA, which filters out the system noise and the noise generated in the correlation process and further

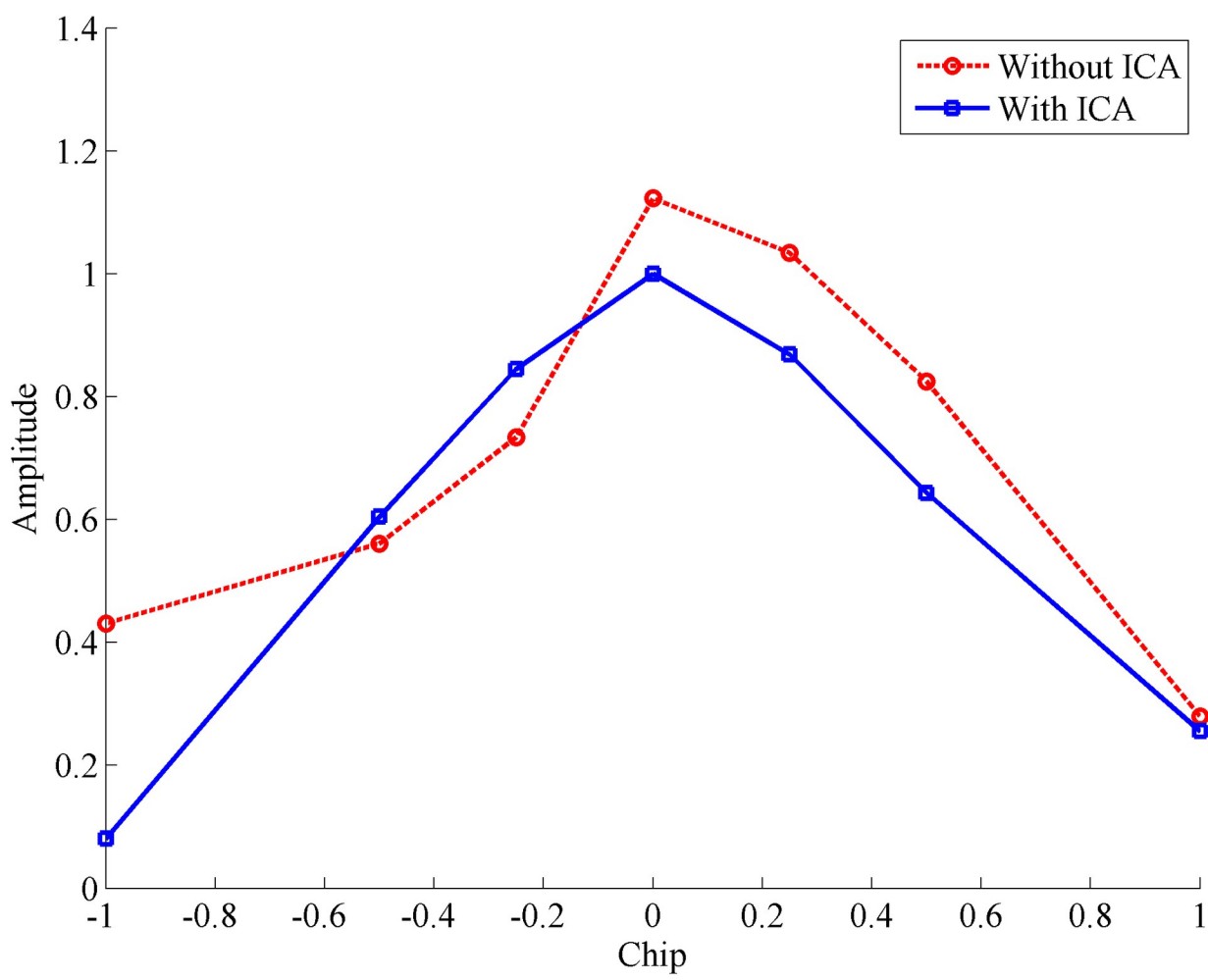

**Fig 3. Comparison of code phase curves.**

improves the SNR of the navigation message after despreading and demodulation. In order to verify the effectiveness and performance of the proposed method, the postcorrelation ICA method is compared with the traditional array signal processing, that is, the spatial smoothing combined with the multiple signal classification (MUSIC) algorithm [32] is used to perform DOA estimation on the received multipath signals, and then the estimated DOAs of direct component and multipath components are used for beamforming processing of the received signals, and finally the processed signal is acquired and tracked, and the navigation message is obtained after despreading.

In terms of computational complexity, the proposed method includes postcorrelation and ICA algorithm. Among them, the postcorrelation and GNSS signal tracking adopt the same algorithm. Therefore, the difference in computational complexity between the proposed method and the traditional method depends on the algorithms adopted to suppress multipath. In traditional array signal processing, the complexity of spatial smoothing is $O\left(M^2L\right)$, the complexity of MUSIC algorithm is $O\left(M^2V\right)$, and the complexity of beamforming algorithm such as Linearly constrained minimum variance (LCMV) is $O\left(M^2L\right)$ [33]. Where $M$ is the number of array elements, $L$ is the number of snapshots and $V$ is the searching grid number of MUSIC algorithm. The total computational complexity of traditional methods is $O\left(M^2L+M^2V\right)$. For

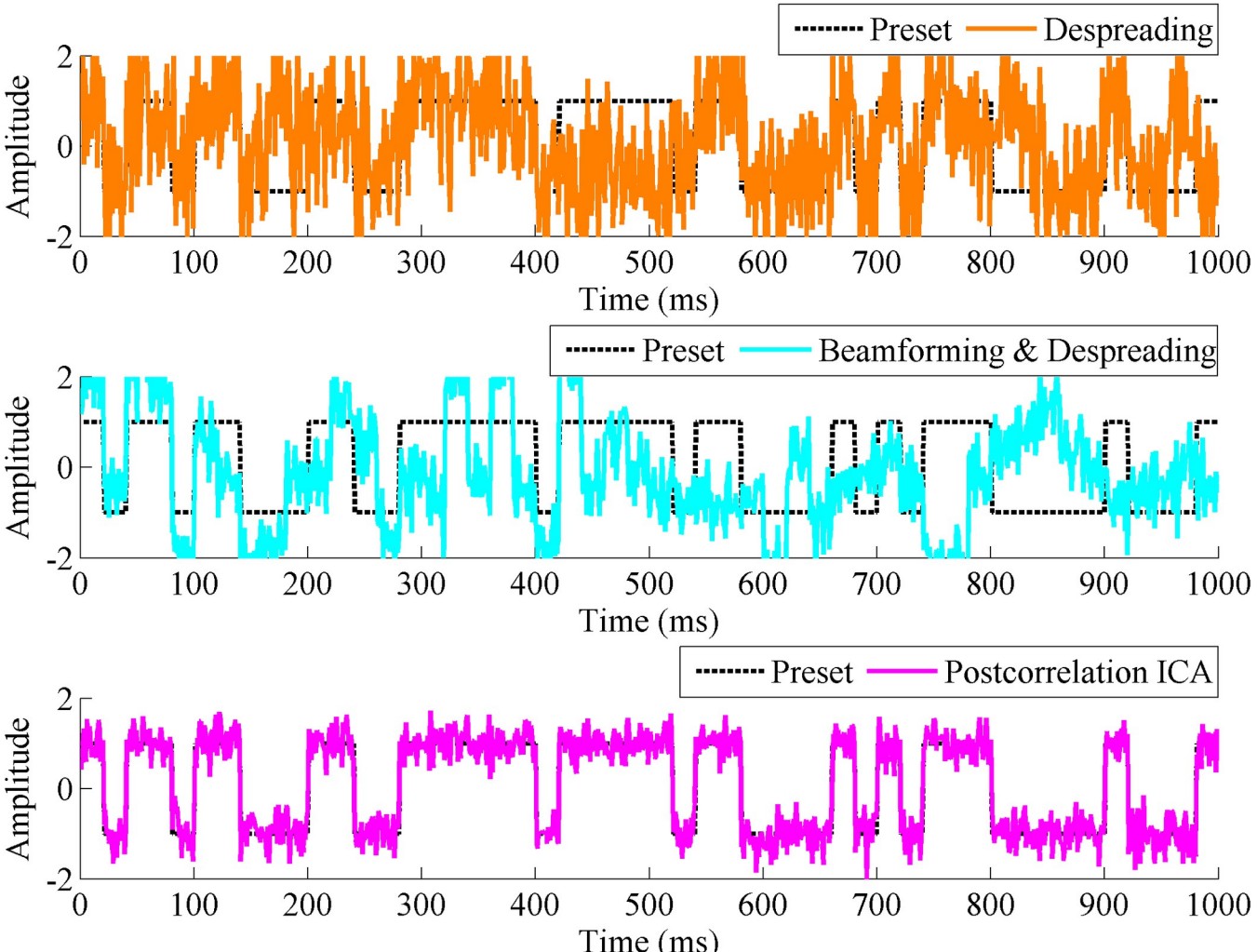

**Fig 4. Comparison of the navigation message after despreading, beamforming and despreading, and postcorrelation ICA algorithm processing.**

the ICA algorithm used in the proposed method, only one direct signal is extracted, so the number of iterations is 1. Therefore, the computational complexity of the proposed method is $O\left(M^2L\right)$, which is smaller than the traditional method.

Fig 4 shows the comparison between the navigation message after dispreading (without postcorrelation ICA), beamforming and despreading, and postcorrelation ICA processing.

Fig 4 shows that when the received SNR is only -30 dB and the correlator is affected by multipath components, the quality of the navigation message obtained after demodulation and despreading is very poor, serious noise and despreading errors occur, and the SNR is approximately -2 dB. After the direct signal is extracted by the ICA algorithm, the quality of the navigation message is significantly improved, the SNR is increased to approximately 11 dB and the bit error rate is zero, because the noise generated in the correlation is filtered out. In the traditional method, due to the low SNR of the received signal, the DOAs of the direct component and the multipath components cannot be accurately estimated, resulting in that the SNR and the bit error rate of the navigation message obtained by beamforming and despreading are not improved.

## 4.3 Verification of the influence of multipath delay on navigation messages

In traditional receivers, the signal quality of navigation messages is affected by multipath interference. In order to study this phenomenon more clearly, the simulation conditions are simplified as only one multipath component existing in the signal of the first satellite, and there is no multipath component in the signals of the remaining two satellites. The multipath incidence angle is 30˚, and the multipath attenuation is 0.2 times the direct signal strength. The delay range varies from 0 to 0.6 chips, and the other conditions are consistent with the general simulation conditions. After every 100 independent simulations, the changes in the SNR of the navigation message with the multipath delay are compared after despreading, beamforming and despreading, and postcorrelation ICA processing, as shown in Fig 5.

As Fig 5 shows, in the case of no multipath delay, the multipath component generates a gain in the signal strength of the direct component to obtain the maximum signal-to-noise ratio of the navigation message. When the multipath delay reaches 0.2 and 0.3 chips, the navigation message is seriously affected by the multipath delay, and the SNR decreases greatly and cannot despread normally. When the delay reaches 0.5 chips, the despreading returns to normal, and the gain generated by the multipath component decreases gradually. The quality of

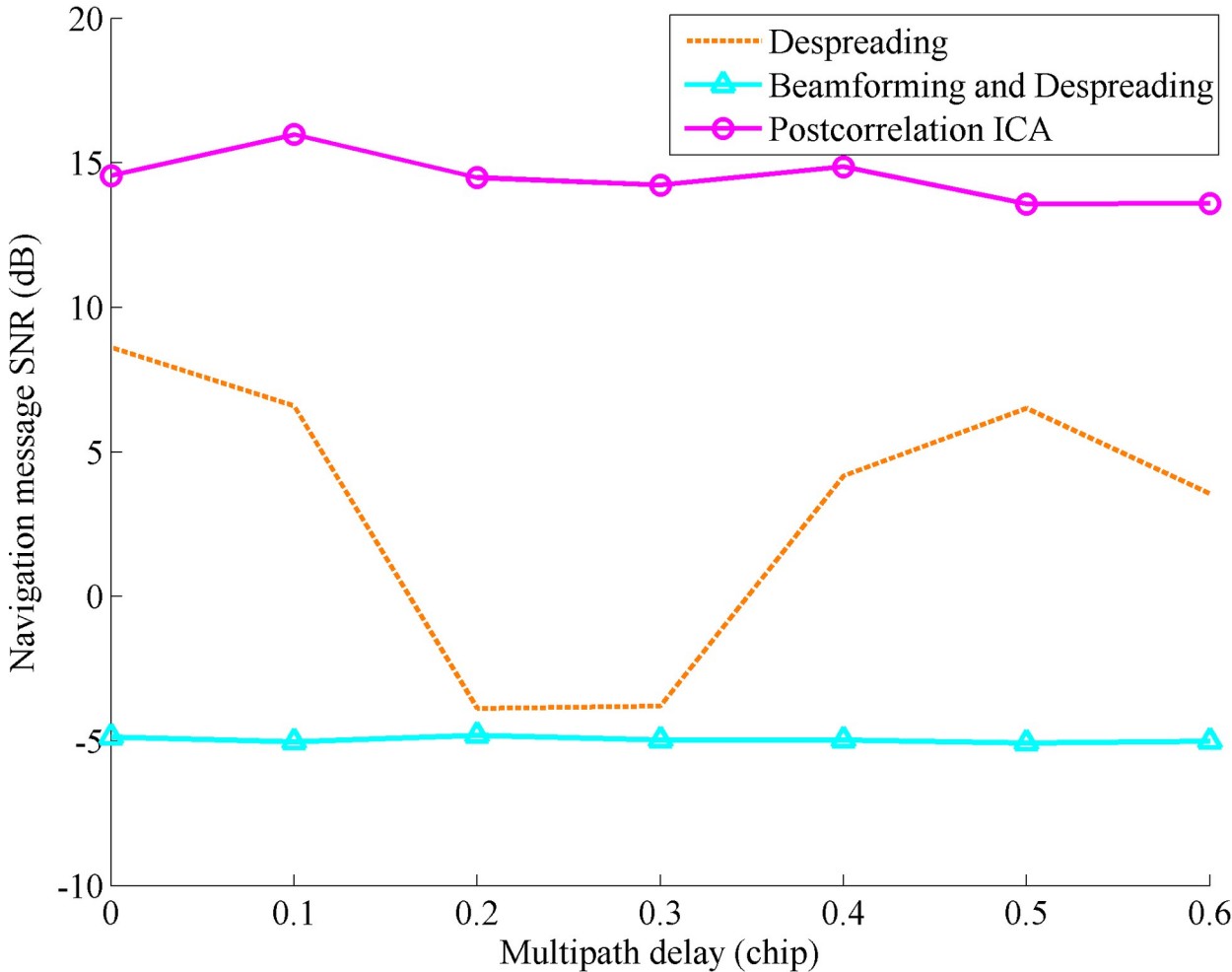

**Fig 5. SNR comparison of the navigation message with the multipath delay changing after despreading, beamforming and despreading, and postcorrelation ICA processing.**

the navigation message extracted after ICA processing is not affected by multipath delay and multipath attenuation. Because some noise is filtered at the same time, the SNR increases to more than 14 dB and remains stable. In the traditional method, due to the low SNR of the received signal, the DOAs of the direct component and the multipath components cannot be estimated accurately, and the SNR of the navigation message obtained by beamforming and despreading has not been improved. In the traditional method, due to the low SNR of the received signal, the SNR of the navigation message obtained by beamforming and despreading does not change with the multipath delay changing.

Fig 6 shows the bit error rate comparison of the navigation message with multipath delay changing after despreading, beamforming and despreading, and postcorrelation ICA processing after 100 independent simulations. The simulation conditions are consistent with those of the previous experiment.

The figure shows that the bit error rate of the navigation message is consistent with the situation reflected by the SNR. When the multipath delay reaches 0.2 or 0.3 chips, the navigation message is seriously affected by the multipath delay, the bit error rate is close to 25%. After recovering the direct signal through ICA processing, the signal quality of the navigation

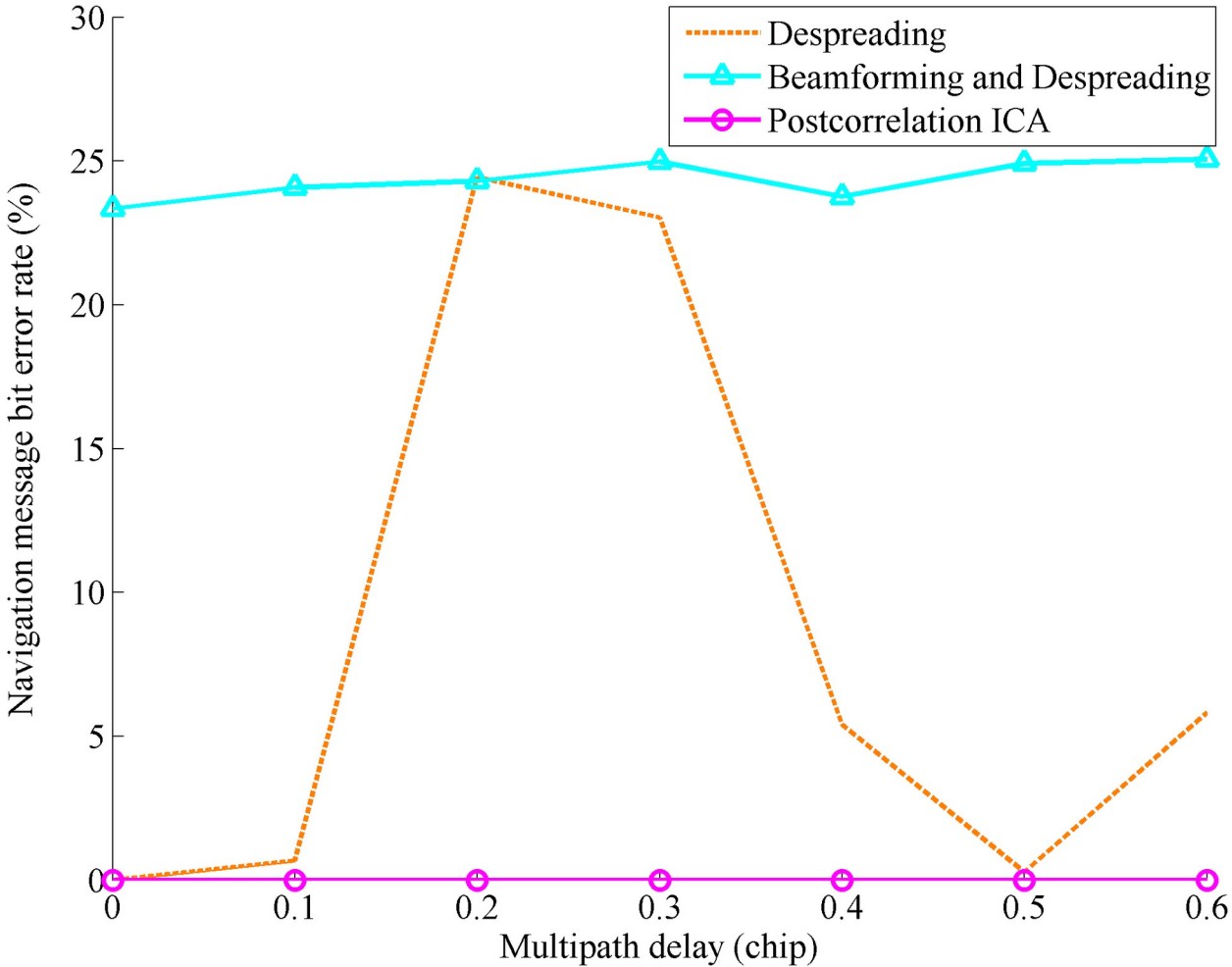

**Fig 6. Bit error rate comparison of the navigation message with the multipath delay changing after despreading, beamforming and despreading, and postcorrelation ICA processing.**

message, which is not affected by multipath delay and multipath attenuation, is greatly improved; and the bit error rate is 0. In the traditional method, due to the low SNR of the received signal, the bit error rate of the navigation message obtained by beamforming and despreading hardly changes with the multipath delay changing.

## 4.4 Verification of the influence of the received SNR on navigation messages

In order to study the influence of the multipath signal on the change of the received SNR and the improvement after using the postcorrelation ICA algorithm, based on the simulation conditions in Section 4.3, the multipath attenuation is set to 0.2 times the direct signal strength, the multipath delay is set to 0.2 chips, and the SNR of the received signal is set to vary from -30 dB to 0 dB. After 100 independent simulations, the changes in the SNR of the navigation message with the received SNR after despreading, beamforming and despreading, and postcorrelation ICA processing are shown in Fig 7.

The simulation results show that the SNR of navigation message affected by multipath signals is still negative after spread spectrum gain when the SNR of the received signal is only -30

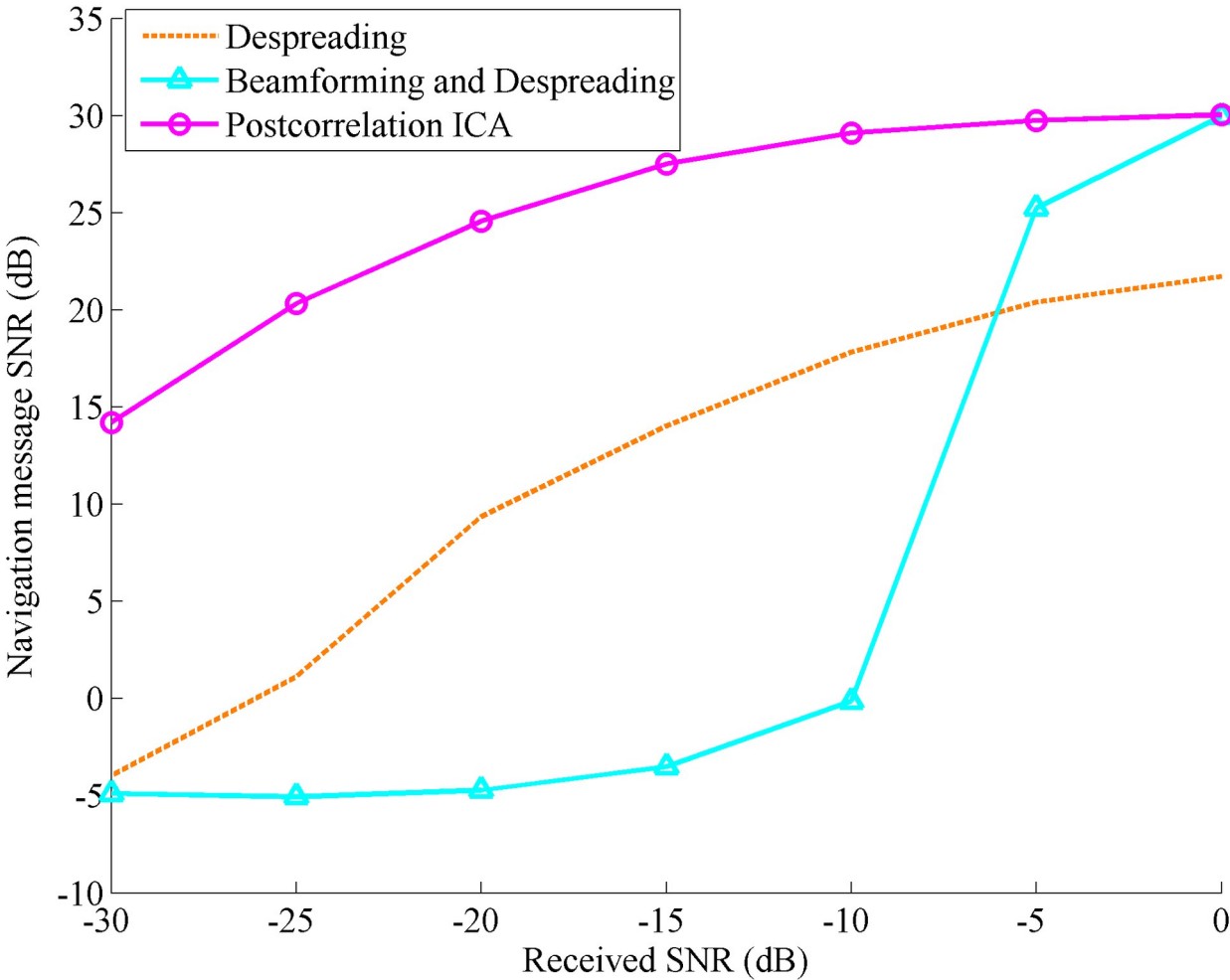

**Fig 7. SNR comparison of the navigation message with the received SNR changing after despreading, beamforming and despreading, and postcorrelation ICA processing.**

dB, but it can reach approximately 14 dB after being processed by the ICA algorithm. As the SNR of the received signal increases, the SNR of the navigation message processed by the ICA algorithm also increases steadily. It can increase by 10 to 20 dB over the entire range of changes, which significantly improves the signal quality. In the traditional method, as the received SNR increases, the accuracy of the estimated DOAs of the direct component and the multipath components is improved, and the SNR of the navigation message after beamforming and despreading is significantly improved. When the received SNR reaches 0 dB, the result is generally consistent with the postcorrelation ICA algorithm.

Fig 8 shows the bit error rate comparison of the navigation message with received SNR changing after despreading, beamforming and despreading, and postcorrelation ICA processing after 100 independent simulations.

The Fig 8 shows that the bit error rate of navigation message affected by multipath signals is close to 25% after spread spectrum gain when the SNR of the received signal is only -30 dB, but it can be reduced to 0 after being processed by the ICA algorithm. With the increase of the received SNR, the bit error rates of the navigation message obtained by dispreading, beamforming and

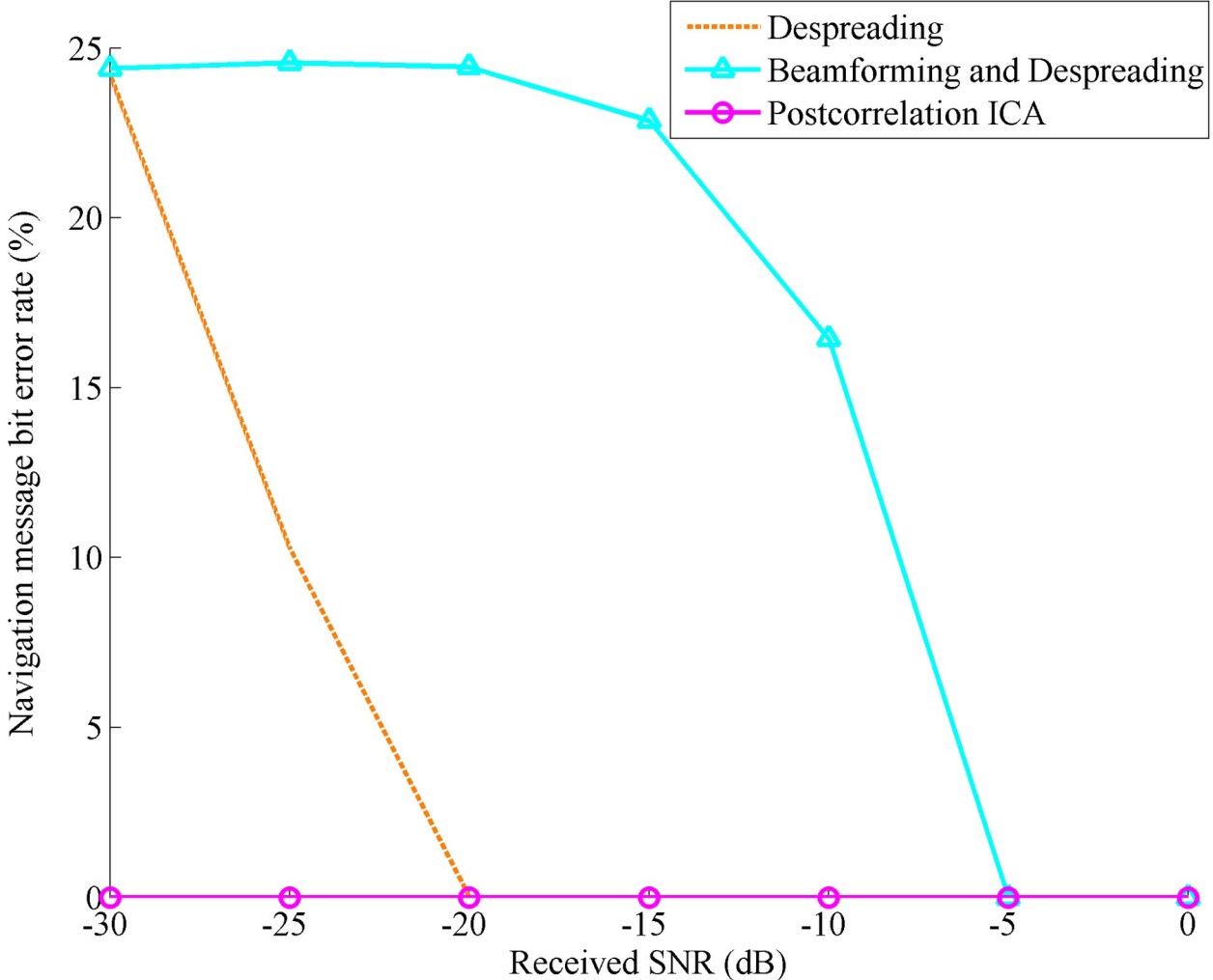

**Fig 8. Bit error rate comparison of the navigation message with the received SNR changing after despreading, beamforming and despreading, and postcorrelation ICA processing.**

despreading are both reduced, but the latter is slower to reduce to 0. When the received SNR reaches -5 dB, the bit error rate of the navigation message obtained by the traditional method reaches 0. It can be seen that compared with the postcorrelation ICA algorithm, the traditional method is actually difficult to use in the urban environment with weak GNSS signal strength.

## 5. Conclusion

This paper proposes a multipath suppression method based on postcorrelation and ICA for satellite navigation signals. Due to the weak strength of satellite navigation signals, it is difficult to first estimate the direction of arrival and then suppress multipath signals. In the proposed method, the ICA algorithm is added to the tracking loop. First, the received signals on all array elements are processed by postcorrelation technology, the spread spectrum gain is used to improve the signal-to-noise ratio, and then the direct signal component is extracted by the ICA algorithm to suppress multipath signals. In addition, the ICA algorithm can also be used to filter noise, and the signal quality of the navigation message is further improved after despreading and demodulation.

The simulation results show that the algorithm can effectively suppress short multipath, medium and long multipath interference when the SNR of the satellite navigation signal is as low as -30 dB and filter out some noise and system noise generated in the correlation process so as to greatly improve the signal quality of the navigation message. The results are not affected by multipath delay and multipath attenuation, and the bit error rate is 0. Within the range where the received SNR changes from -30 dB to 0 dB, the SNR of the navigation message will continue to increase by 10 to 20 dB after demodulation and despreading.

Follow-up research will test the postcorrelation ICA algorithm in actual environments, including the suppression of multipath signals and the improvement of the positioning effect.

## Acknowledgments

Thank Jicheng Ding for providing some ideas and some program codes for this subject.

## Author Contributions

**Conceptualization:** Jian Xu, Jicheng Ding.

**Data curation:** Jian Xu.

**Formal analysis:** Jian Xu, Jicheng Ding.

**Funding acquisition:** Jian Xu.

**Investigation:** Jian Xu, Jicheng Ding.

**Methodology:** Jian Xu, Jicheng Ding.

**Project administration:** Jian Xu.

**Resources:** Jian Xu.

**Software:** Jian Xu, Jicheng Ding.

**Supervision:** Jian Xu.

**Validation:** Jian Xu.

**Visualization:** Jian Xu.

**Writing – original draft:** Jian Xu.

**Writing – review & editing:** Jian Xu.

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
