## [Decision Letter · Decision Letter 0]

10 Dec 2021

PONE-D-21-34788GNSS Multipath Suppression Technology Based on Postcorrelation and Independent Component AnalysisPLOS ONE

Dear Dr. Xu,

Thank you for submitting your manuscript to PLOS ONE. After careful consideration, we feel that it has merit but does not fully meet PLOS ONE’s publication criteria as it currently stands. Therefore, we invite you to submit a revised version of the manuscript that addresses the points raised during the review process.

We look forward to receiving your revised manuscript.

Kind regards,

Chan Hwang See, Ph.D.

Academic Editor

PLOS ONE

Journal Requirements:

Financial support was obtained from Harbin University of Commerce's school-level scientific research project (18XN023).

NO - Include this sentence at the end of your statement: The funders had no role in study design, data collection and analysis, decision to publish, or preparation of the manuscript.

Additional Editor Comments:

Your manuscript entitled "GNSS Multipath Suppression Technology Based on Postcorrelation and Independent Component Analysis" has been reviewed. Reviewers felt that the work is required a major revision. In particular, the novelty/contribution of this work, limited theoretical innovation and required rigorous experimental comparison for verifying the presented results, and clarification of the computational complexity of the proposed method.

Hence, we do encourage you to address the concerns and criticisms of the reviewers detailed at the bottom of this letter and resubmit your article once you have updated it accordingly.

Reviewers' comments:

Reviewer's Responses to Questions

**Comments to the Author**

1. Is the manuscript technically sound, and do the data support the conclusions?

Reviewer #1: Yes

Reviewer #2: Yes

Reviewer #3: Yes

2. Has the statistical analysis been performed appropriately and rigorously? 

Reviewer #1: Yes

Reviewer #2: Yes

Reviewer #3: No

3. Have the authors made all data underlying the findings in their manuscript fully available?

Reviewer #1: Yes

Reviewer #2: Yes

Reviewer #3: No

4. Is the manuscript presented in an intelligible fashion and written in standard English?

Reviewer #1: Yes

Reviewer #2: Yes

Reviewer #3: Yes

5. Review Comments to the Author

Reviewer #1: This paper deals with medium- and short-delay multipath interference suppression problem for satellite navigation signals in complex environment. There are some issues that I can't understand listed here.

1.Line 109, the explanations of the correlator is not clear.

2.Although overall readable, there exists some smallgrammar errors.

Reviewer #2: This paper proposed a multipath suppression method based on postcorrelation and ICA for satellite navigation signals. Simulation results shown that the proposed method can effectively suppress medium and short multipath interference when the SNR of satellite navigation signals is low, significantly improve the SNR of navigation messages and reduce the bit error rate, which provides a new solution for global navigation satellite system (GNSS) multipath suppression.

There are several comments as below.

1. The authors are suggested to include the rationality of multipath setting in existing simulation experiments.

2. How about the computational complexity of the proposed method?

3. How about the influence of different kinds of noise?

4. The authors are suggested to compare the proposed method with traditional methods.

Reviewer #3: The manuscript combines some mature theories in an attempt to solve the multipath in urban environment, but the language of the manuscript needs to be further revised, and more comparative experiments need to be added to support the effectiveness of the method

6. PLOS authors have the option to publish the peer review history of their article (what does this mean?). If published, this will include your full peer review and any attached files.

Reviewer #1: No

Reviewer #2: No

Reviewer #3: No

---

## [Author Response · Author response to Decision Letter 0]

8 Feb 2022

Responses to Reviewers

Reviewer #1:

Comments 1: Line 109, the explanations of the correlator is not clear.

Response: 

Thanks for your comments. Like the correlator in an ordinary receiver, its function is to despread the received signal, peel off the carrier and get the navigation message. If the array received signals are input into the correlator to obtain the navigation messages, it is equivalent to using an array antenna to directly receive navigation messages containing multipath signals, which can be described as equation (3) in line 109.

The description in the manuscript has been revised.

Comments 2: Although overall readable, there exists some smallgrammar errors.

Response: 

The grammar errors in the manuscript have been copyedited by American Journal Experts. The remaining small grammamar errors will also be corrected as far as possible.

Reviewer #2:

Comments 1: The authors are suggested to include the rationality of multipath setting in existing simulation experiments.

Response: 

Thanks for your comments. In the urban environment, the multipath received signals come from different obstacles, and these signals have different amplitudes and phases. The overall distribution of the final synthetic signal subject to Rayleigh distribution or Rice distribution according to whether there is a direct signal. The multipath signal simulation parameters set in this paper subject to Rice distribution, and the relevant descriptions, simulation and references have been added in the revised manuscript.

Comments 2: How about the computational complexity of the proposed method?

Response: 

The proposed method includes two parts, post-correlation (GNSS signal tracking) and ICA algorithm. Among them, GNSS signal tracking is the operation of satellite navigation receiver itself, so the computational complexity of the proposed method only comes from ICA algorithm. Therefore, the computational complexity is O(n).

Comments 3: How about the influence of different kinds of noise?

Response: 

The simulation of GNSS receiver background noise usually adopts additive white Gaussian noise, so the impact of other kinds of noise on the algorithm is not analyzed in the manuscript. We replace white Gaussian noise with flicker noise (pink noise), and get better simulation results than before (Not listed in the revised manuscript). As shown in the figure below, when the received signal-to-noise ratio is - 30dB, it rises to 19dB after processing by the proposed method. Except for the noise type, the simulation parameters are consistent with those in the manuscript.

Comments 4: The authors are suggested to compare the proposed method with traditional methods.

Response: 

We have compared the proposed method with traditional methods in revised manuscript. 

In the traditional method, the DOAs of received multipath signal is estimated by spatial smoothing combined with the MUSIC algorithm, then estimated DOAs of direct component and multipath components are used to beamform the received signal, and finally the processed signal is tracked to obtain the navigation messages. By comparing the signal-to-noise ratio and bit error rate of navigation messages obtained by the traditional method and the proposed method, the advantages of the new method can be presented.

The description in the manuscript has been revised.

Reviewer #3:

Comments 1: The multipath received in urban environment is not only the reflection signal but also diffraction, refraction etc. Please give more rigorous definition in your first paragraph of your introduction and give the reference.

Response: 

Thanks for your comments. In urban environment, reflection, diffraction and scattering on the building surfaces in the radio environment induce undesirable multipath propagation. The relevant descriptions and references have been added in the revised manuscript.

Comments 2: Line 62: For “DOA”, please give the full spelling when you first introduce it.

Response: 

The description in the manuscript has been revised.

Comments 3: Actually, there are many new methods (Neural Network, machine learning) to recognize line of sight and non-line of sight in the process of signal acquisition and tracking. You should not ignore them in your introduction.

Response: 

The relevant descriptions and references have been added in the revised manuscript.

Comments 4: Your introduction is well-written on the whole, but most of your references are too old and some references for multipath processing need to be added to the review of the multipath mitigation. In particular, the current research on the spatio-temporal correlation of multipath errors (i.e. SF, ASF, MHM, T-MHM, SD-MHM). Moreover, you should cite some references about ICA for multipath mitigation (i.e. https://link.springer.com/article/10.1007/s10291-013-0341-9) or similar to ICA method (i.e. https://link.springer.com/article/10.1007/s10291-020-01074-y). These studies should be cited but not limited to these.

Response: 

The relevant descriptions and references have been added in the revised manuscript.

Comments 5: You spend a lot of time introducing ICA. Please explain, ICA methods have two main problems. The component amplitude is not uniform and the signal source is uncertain. How did you solve or avoid? Please explain or modify in detail.

Response: 

In the proposed method, there is only a single satellite signal on each tracking loop, the number of source signals is 1, so the navigation message is extracted directly by ICA. For the first problem, since the output result of the proposed method is navigation message, the signal amplitude can be standardized. For the second problem, the signal source is confirmed by acquisition processing.

Comments 6: The experiment is too simple to simulate the Beidou navigation signals. You should add smaller multipath delay time to verify the feasibility.

Response: 

The multipath delay time is shortened in the revised manuscript.

Comments 7: I have no idea which generation of Beidou system the author simulated. Please give more detailed simulation parameters for your Beidou signals in table 1.

Response: 

The signal of the second generation Beidou system is used in the simulation. This parameter is added in Table 1 in the revised manuscript.

Comments 8: It is better to give a comparative experiment in Fig 3. I think most of the filtering methods can play a good role after whitening.

Response: 

Previously, whitening was added in the verification of navigation message effectiveness to show the implementation process of the proposed method. After consideration, it seems unnecessary, or it is more meaningful to add the comparison with the traditional algorithm. Therefore, whitening has been deleted from the comparison chart in the revised manuscript.

Comments 9: Please explain, what are the conditions for the use of the proposed method? Is it more suitable in a dynamic environment, or does it perform better in a static environment? Furthermore, it is strongly recommended to add a comparison method to reflect the advantages of the proposed method.

Response: 

The proposed method is suitable for both dynamic and static environments, because the essence of the method is to add ICA processing in the signal tracking process of GNSS receiver, which will not have much impact on the positioning time of the receiver.

In the revised manuscript, the proposed method has been compared with the traditional beamforming multipath suppression method.

Comments 10: As we all know, although SNR is one of the important properties of GNSS signal, but how does the proposed method perform on PNT after suppressing multipath? The suggestion is based on a carrier to reflect the universality of the proposed method. For example, after suppressing multipath, positioning performance can be demonstrated.

Response: 

The proposed method is to suppress the influence of multipath by improving the navigation message. That is, under the simulation conditions in this paper, the navigation message obtained after despreading is consistent with that without multipath. The improvement of other effects caused by multipath on PNT performance is not within the scope of this study.

---

## [Decision Letter · Decision Letter 1]

22 Mar 2022

PONE-D-21-34788R1GNSS Multipath Suppression Technology Based on Postcorrelation and Independent Component AnalysisPLOS ONE

Dear Dr. Xu,

Thank you for submitting your manuscript to PLOS ONE. After careful consideration, we feel that it has merit but does not fully meet PLOS ONE’s publication criteria as it currently stands. Therefore, we invite you to submit a revised version of the manuscript that addresses the points raised during the review process.

We look forward to receiving your revised manuscript.

Kind regards,

Chan Hwang See, Ph.D.

Academic Editor

PLOS ONE

Journal Requirements:

Additional Editor Comments (if provided):

After the first round of revision, reviewers felt that the work is still required a minor revision. Therefore, authors should carefully address and implement the comments from the reviewers.

The following are some other comments, authors should consider. In particular the following two remarks by the reviewer2.

1. Please include the computational complexities of the proposed method and these traditional methods.

2. Meanwhile, the results of the performance under different SNRs is also need to be included.

Reviewers' comments:

Reviewer's Responses to Questions

**Comments to the Author**

1. If the authors have adequately addressed your comments raised in a previous round of review and you feel that this manuscript is now acceptable for publication, you may indicate that here to bypass the “Comments to the Author” section, enter your conflict of interest statement in the “Confidential to Editor” section, and submit your "Accept" recommendation.

Reviewer #2: All comments have been addressed

Reviewer #3: All comments have been addressed

2. Is the manuscript technically sound, and do the data support the conclusions?

Reviewer #2: Yes

Reviewer #3: Yes

3. Has the statistical analysis been performed appropriately and rigorously? 

Reviewer #2: Yes

Reviewer #3: I Don't Know

4. Have the authors made all data underlying the findings in their manuscript fully available?

Reviewer #2: Yes

Reviewer #3: No

5. Is the manuscript presented in an intelligible fashion and written in standard English?

Reviewer #2: Yes

Reviewer #3: Yes

6. Review Comments to the Author

Reviewer #2: Some minor comments.

1. Please include the computational complexities of the proposed method and these traditional methods.

2. Meanwhile, the results of the performance under different SNRs is also need included.

Reviewer #3: This article still does some work for multipath mitigation, I hope the author can do more in this field

7. PLOS authors have the option to publish the peer review history of their article (what does this mean?). If published, this will include your full peer review and any attached files.

Reviewer #2: No

Reviewer #3: No

---

## [Author Response · Author response to Decision Letter 1]

2 Apr 2022

Reviewer #2:

Comments 1: Please include the computational complexities of the proposed method and these traditional methods.

Response: 

Thanks for your comments. The proposed method includes two parts, postcorrelation and ICA algorithm. Among them, the postcorrelation and GNSS signal tracking adopt the same algorithm. Therefore, the difference in computational complexity between the proposed method and the traditional method depends on the algorithms adopted to suppress multipath. In traditional array signal processing, the complexity of spatial smoothing is O (M2L), the complexity of MUSIC algorithm is O (M2V), and the complexity of beamforming algorithm such as LCMV is O (M2L). Where M is the number of array elements, L is the number of snapshots and V is the searching grid number of MUSIC algorithm. The total computational complexity of traditional methods is O (M2L+M2V). For the ICA algorithm used in the proposed method, only one direct signal is extracted, so the number of iterations is 1. Therefore, the computational complexity of the proposed method is O (M2L), which is smaller than the traditional method.

The description in the manuscript has been revised.

Comments 2: Meanwhile, the results of the performance under different SNRs is also need included.

Response: 

Section 4.4 shows the performance comparison between the proposed method and the traditional method under different received SNR.

Reviewer #3:

Comments 1: This article still does some work for multipath mitigation, I hope the author can do more in this field.

Response: 

Thank you for your previous valuable suggestions on improving this article. We will conduct more in-depth research in the field of multipath mitigation.

---

## [Editor Report · Decision Letter 2]

5 Apr 2022

GNSS Multipath Suppression Technology Based on Postcorrelation and Independent Component Analysis

PONE-D-21-34788R2

Dear Dr. Xu,

We’re pleased to inform you that your manuscript has been judged scientifically suitable for publication and will be formally accepted for publication once it meets all outstanding technical requirements.

Kind regards,

Chan Hwang See, Ph.D.

Academic Editor

PLOS ONE
---

## [Editor Report · Acceptance letter]

8 Apr 2022

PONE-D-21-34788R2 

GNSS Multipath Suppression Technology Based on Postcorrelation and Independent Component Analysis 

Dear Dr. Xu:

I'm pleased to inform you that your manuscript has been deemed suitable for publication in PLOS ONE. Congratulations! Your manuscript is now with our production department. 

Kind regards, 

on behalf of

Dr. Chan Hwang See 

Academic Editor

PLOS ONE